# Investigation on Antioxidant Activity and Different Metabolites of Mulberry (*Morus* spp.) Leaves Depending on the Harvest Months by UPLC–Q-TOF-MS with Multivariate Tools

**DOI:** 10.3390/molecules28041947

**Published:** 2023-02-17

**Authors:** Zili Guo, Jiangxuan Lai, Yiwen Wu, Sheng Fang, Xianrui Liang

**Affiliations:** 1Key Laboratory for Green Pharmaceutical Technologies and Related Equipment of Ministry of Education, College of Pharmaceutical Sciences, Zhejiang University of Technology, Hangzhou 310000, China; 2School of Food Science and Biotechnology, Zhejiang Gongshang University, Hangzhou 310018, China

**Keywords:** mulberry leaf, chemical constituent, antioxidant, UPLC–Q-TOF-MS, free radical scavenging activity

## Abstract

The changes in active components in mulberry leaves harvested in different months and their antioxidant activities were investigated. Ultra-high-performance liquid chromatography–tandem quadrupole time-of-flight mass spectrometry (UPLC–Q-TOF-MS) with multivariate statistical tools was used to investigate the chemical constituents in the extracts of mulberry leaves. The results indicated that mulberry leaves were rich in phenolic acids, flavonoids, organic acids, and fatty acid derivatives. In addition, 25 different compounds were identified in the different batches of mulberry leaves. The 1,1-diphenyl-2-picrylhydrazyl (DPPH) radical scavenging activity was measured to evaluate the in vitro antioxidant activities of mulberry leaves. Among the four batches, batch A, harvested in December, exhibited the strongest DPPH radical-scavenging activity, while batch B, harvested in March, showed the weakest activity. This was related to the total phenolic content in the mulberry leaves of each batch. The optimal harvest time of mulberry leaves greatly influences the bioactivity and bioavailability of the plant.

## 1. Introduction

Mulberry (*Morus* spp.) has been widely cultivated in many Asian countries, such as China, India, Korea, Japan, and Thailand. Mulberry leaves, commonly used as food for silkworms in sericulture, have been commercially available as a kind of special tea or drink in many Asian countries [1,2,3]. However, mulberry leaves are also traditionally applied as folk medicine to treat fever, protect the liver, improve eyesight, strengthen the joints, facilitate the discharge of urine, and lower blood pressure [4,5,6]. Modern pharmacological research has revealed that mulberry leaves also have a broad range of biological activities, such as antioxidant [7,8], anti-inflammatory [9,10,11], anti-bacterial [12], anti-hypertensive [13], anti-atherogenic [14], and anti-cancer [15] effects. Studies of the chemical constituents of mulberry leaves have shown that mulberry leaves are rich in phenols, flavonoids, alkaloids, amino acids, polysaccharides, and steroids [16,17,18,19,20]. The species and content of secondary metabolites may vary with the different growth stages, which results in potentially different pharmacological activities. The literature has reported that the degree of maturity and the harvest time of mulberry leaves significantly affected the content of nutritional and functional components [21,22,23,24].

The phenols have been reported to be one of the main components in mulberry leaves. The phenolic content in mulberry leaves has been found to be greatly influenced by the leaf age (tips, young, and old leaves) [23] and seasonal changes [24]. The tips of the leaves were taken from positions 1 to 3 from the top of each branch; young leaves were taken from positions 4 to 6; and old leaves were taken from positions 7 to 10. The results showed that the phenolic content in the tips of leaves was higher [23]. The in vitro antioxidant capacity of different-aged mulberry leaves was detected and compared with clear superoxide radical (O_2_^−•^), DPPH free radical scavenging, hydroxyl radical scavenging, and Fe^2+^ chelating and reducing activities, respectively [25,26]. Mulberry leaves collected in May were considered to be preferred because of their higher phenolic content [21,22]. There were differences in the levels of the seven phenolic compounds (chlorogenic acid, benzoic acid, rutin, isoquercitrin, astragalin, quercetin-3-*O*-(6-*O*-malonyl)-β-D-glucoside, and kaempferol-3-*O*-(6-*O*-malonyl)-β-D-glucoside) during the growing seasons, with leaves collected from April to October. However, the literature only reported the content changes in a very few polyphenolic compounds in mulberry leaves, and the studies were relatively limited. The quantitative changes in other functional constituents in mulberry leaves need to be clarified. 

Thus, the quantitative changes in functional constituents in mulberry leaves according to the harvest month were studied. In the Huzhou area (a traditional sericulture area), March, April, May, and December represent the growth periods and different picking periods for mulberry leaves. Mulberry leaves have different uses in different periods; for example, the young shoots of mulberry leaves can be used for cooking in March, big mulberry leaves are plucked in April and May to feed silkworms, and old mulberry leaves are picked after the frost for herbal tea in December. Therefore, the mulberry leaves collected in March, April, May, and December were selected and studied using an ultra-high-performance liquid chromatography–quadrupole time-of-flight mass spectrometry (UPLC–Q-TOF-MS) technique. The UPLC–Q-TOF-MS technology has the required sensitivity for fast, high-resolution separations and also facilitates the structure elucidation and identification of fragmentation patterns [27]. The antioxidant activities of mulberry leaves at different harvest times were also evaluated using the DPPH assay. Meanwhile, the total phenolic content of mulberry leaves in each batch was measured. An investigation of the influence of mulberry leaves harvested in different months on these variables would be helpful to understand the health benefits of mulberry leaves.

## 2. Results and Discussion

### 2.1. Optimization of Extraction Conditions

Taking the main seven compounds in mulberry leaves as target compounds (as shown in Appendix A), the solvent extraction conditions of mulberry leaves were optimized by single-factor experiments. The extraction conditions, including extraction solvents (Water, N-hexane, ethyl acetate, methanol, ethanol, 15% ethanol solution, 30% ethanol solution, 45% ethanol solution, 60% ethanol solution, 75% ethanol solution, and 90% ethanol solution), extraction methods (maceration, heat-reflux extraction, and ultrasonic extraction), liquid/solid ratios (10–40 mL/g), extraction times (20–50 min), and extraction temperatures (25–45 °C), were all tested and the results are shown in Appendix A, respectively. According to the number, separation, and areas of the peaks, the preferred extraction conditions include an extraction solvent of 75% ethanol solution, an extraction method of ultrasonic extraction, a liquid/solid ratio of 30 mL/g, an extraction time of 30 min, and an extraction temperature of 35 °C. 

### 2.2. Optimization of UPLC and MS Conditions

Various UPLC parameters, including columns, mobile phases, and column temperature, were evaluated to achieve efficient separation, a better peak shape, and a reasonable analysis time. The columns (BEH C18, 100 × 2.1 mm, 1.7 µm; HSS T3, 50 × 2.1 mm, 1.8 µm; and HSS T3, 100 × 2.1 mm, 1.8 µm), mobile phases (methanol–water, acetonitrile–water, acetonitrile–0.1% formic acid aqueous solution, and acetonitrile–0.2% formic acid aqueous solution), column temperatures (25, 30, 35, and 40 °C), flow rates (0.2, 0.25, and 0.3 mL/min), and monitor wavelengths (254, 280, and 320 nm), were all tested. The preferred column was the HSS T3 column (100 × 2.1 mm, 1.8 µm). The mobile phase was an acetonitrile–0.1% formic acid aqueous solution. The ultimate flow rate was 0.2 mL/min with the column temperature at 30 °C. The optimal monitor wavelength was 254 nm.

To acquire the maximum sensitivity for most constituents, the Q-TOF-MS and Q-TOF-MS/MS parameters were also optimized in both positive and negative modes. The drying gas was run at various flow rates (6.0, 8.0, and 10.0 L/min) and temperatures (180, 200, and 220 °C). The optimum conditions were determined as follows: The capillary voltage was 3800 V in the negative mode; the dry gas flow rate was 6.0 L/min; the nebulizer gas pressure was 0.8 bar; and the dry gas temperature was set at 200 °C.

### 2.3. Identification of the Compounds by UPLC–Q-TOF-MS

UPLC–Q-TOF-MS coupled with an electrospray ionization (ESI) ion source in the negative mode was performed to analyze the compounds of mulberry leaf extracts using 75% ethanol extracts of batch B. The base peak chromatogram (BPC) of mulberry leaf extracts is shown in Figure 1. The information from UPLC–Q-TOF-MS data and the names of speculated compounds are summarized in Table 1.

As is shown in Table 1 and Figure 1, a total of 77 compounds were detected based on retention time (RT), exact mass data, fragment information, and molecular formulas reported in the literature, and 65 chemical structures were tentatively deduced and identified. Among them, there were 5 organic acids, 1 vitamin derivative, 15 phenolic compounds, 31 flavonoids, 6 fatty acid derivatives, 3 terpenoids, 2 quinonoids, 1 lignan, and 1 terpene. Flavonoid glycosides were the main compounds found in the 75% ethanol extracts of mulberry leaves.

Compounds **1**–**6** were identified as organic acids and vitamin derivatives by comparison of their RT, accurate molecular ions, and characteristic fragment ions with those reported in the literature or by the MS-DIAL database. Compounds **61**, **63**, **65**, **70**, **75**, and **77** were identified as fatty acid derivatives by comparison with RT, accurate molecular ions, and characteristic fragment ions, as mentioned in the literature [28]. 

Compounds **7**–**18**, **22**, **23**, and **64** were identified as phenolic compounds. In the MS spectra, all of these compounds showed similar fragmentation pathways by losing a glucose substituent (162 Da) from the precursor ions, and the continuous losses of H_2_O and CO_2_ from the fragment ions. 

Similarly, compounds **21**, **24**, **31**, **32**, **37**–**40**, **43**–**50**, and **54**–**56** were also tentatively identified according to the accurate molecular formulas, the fragmentation pathways, the reference substances, and the reported literature [27,28,29,30,33,35]. Most of these compounds contain a common 15-carbon polyphenolic skeleton and glycosides and are easily deglycosylated to lose the glucose units during MS fragmentation [45,46].

### 2.4. Investigation of the Differential Chemical Constituents of Mulberry Leaves Harvested in Different Months

#### 2.4.1. Principal Component Analysis (PCA)

PCA analysis is a commonly used unsupervised discriminant analysis method that can reduce the dimensional display of multi-dimensional data. The projection points are obtained to determine the position of this group of data by projecting the scores of each variable in a group of data onto the principal components. PCA analysis can display multi-dimensional information in a two-dimensional way. Samples gather or separate based on their differences: similar samples will gather, and different samples will separate from each other. QC samples can be used for standardization. They can be used to simulate the difference in signals in the data acquisition process and correct the error of the instrument. The QC data can be used as a training set to establish a prediction model.

The PCA score chart of four batches of mulberry leaves harvested in different months is shown in Figure 2. The model was excellent, with the goodness-of-fit parameter R^2^X (97.6%) and the predicted fitting parameter Q^2^ (96.5%) according to the four principal components. Quality control (QC) samples were gathered near the origin, proving that the experimental operation error and instrument detection error had little influence during the experiment. The intra-group data of each batch of mulberry leaves was well aggregated. The mulberry leaves harvested in different months were well separated between groups, and there were no obvious abnormal points. Among them, mulberry leaves collected in December 2019 (batch A) were alone at the extreme edge of the third quadrant, which was separated from the other three batches. The other three batches of samples were mainly in the first and fourth quadrants, and these samples were collected in the spring and summer of 2020. The PCA result plot was the same as the actual predicted result. It was feasible to investigate the compositional differences in mulberry leaves harvested in different months in an unsupervised manner.

#### 2.4.2. Comparisons of Different Batches of Mulberry Leaves by Orthogonal Partial Least Squares Discriminant Analysis (OPLS-DA)

To further explore the differential compounds among the mulberry leaves harvested in different months, a pairwise comparison of OPLS-DA was carried out. As is shown in Figure 3, a volcano plot combined with the VIP value was used to analyze the differential substances. The results are shown in Table 2.

The purple mark is the differential compound with VIP = 2–3, and the difference was significant. The blue points are differential compounds with VIP > 3, which showed very high differences. The size of the VIP value was reflected by the size of each mark. It showed that the mulberry leaves of batch A showed many different metabolites from batches B, C, and D, which also corresponded to the harvest time. It also proved that the substances in the mulberry leaves harvested in December were very different from those in the other mulberry leaves. Similarly, since the harvest time interval was only one month, there were nearly no significant differences in batches B, C, and D. The results are shown in Table 2.

By comparison, it was found that there were significant differences in the types and content of compounds between batch A and the other three batches. There was little difference between batches B, C, and D. The contents of compounds **40**, **32**, **69**, **66**, **72**, **18**, **37**, **30**, and **10** were different between batches A and B. Among them, compounds **40**, **32**, **69**, **66**, and **72** were significantly different between batches A and B. Compounds **40**, **32**, **69**, and **72** were found in relatively high amounts in batch A; the same was true for compound **66** in batch B. There were more different compounds between A and C/D, including compounds **69**, **66**, **72**, **18**, **37**, **30**, **60**, **59**, **38**, **64**, **58**, **65**, **39**, **8**, **51**, **53**, **40**, and **10**. Compounds **69**, **72**, **37**, **60**, **59**, **38**, and **40** had a higher content in batch A, while compound **65** had a higher content in batch C/D. Since the harvest time interval was only one month, there were nearly no significant differences in batches B, C, and D. Table 2 also shows that the differential compounds were mainly fatty acid derivatives and flavonoids.

### 2.5. DPPH Assay and Assays for Total Phenolics

The DPPH is a stable free radical, which is reduced to α,α-diphenyl-β-picrylhydrazine by reacting with an antioxidant. Antioxidants interrupt free radical chain oxidation by donating hydrogen from hydroxyl groups to form a stable end-product that does not initiate or propagate further oxidation of lipids [16,47]. The results of the free radical-scavenging activity of different mulberry leaves are shown in Figure 4. All the extracts demonstrated significant inhibitory activity against the DPPH radicals. The IC_50_ of batches A, B, C, and D were 0.057, 0.176, 0.162, and 0.090 mg/mL, respectively.

The results of the DPPH scavenging activity of mulberry leaves in different harvest months showed that the free-radical-scavenging activity of mulberry leaves in batch A was the strongest. The free radical scavenging activity in batches C and D was similar, while batch B was the weakest. This showed a certain regularity in the growth period. Batch A of mulberry leaves after frost exhibited the strongest free-radical-scavenging activity. Among the three batches B, C, and D of mulberry leaves, the free-radical-scavenging activity increased with the growth period. Combined with the results of the differential component analysis, this showed that the mulberry leaves of batch A were quite different from batches B, C, and D. The difference was mainly in flavonoid glycosides and simple polyphenols. It was speculated that the free-radical-scavenging activity increases along with the increase in polyphenols during the growth of mulberry leaves.

Based on the absorbance values of the various extract solutions, they were reacted with the Folin–Ciocalteu reagent and compared with the standard solutions of gallic acid. The total phenolic content of batch A was 57.10 mg/g, while that of batches B, C, and D was 35.69, 38.05, and 52.19 mg/g, respectively. Data obtained from the total phenolic assay supported the key role of phenolic compounds in the free radical scavenging of DPPH. As expected, the amount of total phenolics was highest in batch A, harvested in December. The results were consistent with those reported in the previous literature [24]. In the study of [24], the phenolic content in mulberry leaves was high from late May to early July. From late September, the phenolic content increased with time and reached its highest level on October 16 (the last day of the experimental period).

## 3. Materials and Methods

### 3.1. Materials and Chemicals

The mulberry leaves in this experiment were collected from Deqing (119°97′ E, 30°53′ N), Huzhou City, Zhejiang Province, and were authenticated by Dr. Chu Chu (Zhejiang University of Technology, Hangzhou, China). The harvest time is shown in Table 3. After being freeze-dried, the mulberry leaves were stored in a refrigerator at −20 °C before analysis. Chlorogenic acid (≥98%), rutin (≥98%), and isoquercitrin (≥98%) were purchased from Shanghai Yuanye Bio-Technology Co., Ltd. (Shanghai, China). Gallic acid (≥99%) and Folin–Ciocalteu reagent were purchased from Beijing Zhongkezhijian Biotechnology Co., Ltd. (Beijing, China). 

HPLC-grade acetonitrile and methanol were supplied by Merck (Darmstadt, Germany). HPLC-grade formic acid was purchased from Shanghai Aladdin Bio-Chem Technology Co., Ltd. (Shanghai, China). Ultrapure water (18.2 MΩ) was purified using the Milli-Q^®^ IQ 7000 Purification System (Molsheim, France). Other reagents used in this experiment were all of analytical grade and were obtained from Yongda Chemical Reagent Company (Tianjin, China).

### 3.2. Sample Preparation

#### 3.2.1. Preparation of Different Batches of Samples

Four batches of dried mulberry leaves harvested in different months were frozen overnight in an ultra-low-temperature refrigerator and freeze-dried to achieve a constant weight. They were fully ground and passed through an 80-mesh sieve. The dried powder (1.0 g) was extracted with a 75% ethanol solution (1:30, *w*/*v*) for half an hour in an ultrasonic bath at 35 °C. Next, the supernatant filtered from the total extract was concentrated by decompression and evaporation. The residue obtained was then dissolved in a 10 mL 80% methanol solution and filtered through a 0.22 µm PTFE membrane before the UPLC–Q-TOF-MS analysis.

#### 3.2.2. Preparation of QC Samples

QC samples were prepared by mixing aliquots of four batches of samples to form a pooled sample, and they were then analyzed in the same way as the analytic samples [48]. 

### 3.3. UPLC–Q-TOF-MS Conditions

The chromatographic separation experiment was carried out using an Ultimate^TM^ 3000 UPLC system (Thermo Scientific, DIONEX, Sunnyvale, CA, USA) equipped with an RS pump, an RS autosampler, an RS column compartment, an RS variable wavelength detector, and a compact mass spectrometer (Bruker Daltonics, Bremen, Germany) using composite ESI in the negative ion mode. The separation was operated on a Waters Acquity UPLC HSS T3 column (100 × 2.1 mm, 1.8 µm, Waters). The mobile phase consisted of 0.1% formic acid aqueous solution (*v*/*v*) (A) and acetonitrile (B) with gradients of 0–4 min, 2–7% B; 4–7 min, 7–10% B; 7–12 min, 10–12% B; 12–20 min, 12–22% B; 20–25 min, 22–35% B; 25–28 min, 35–50% B; 28–31 min, 50–70% B, 31–37 min, 70–90% B, 37–45 min, 90–2% B. The flow rate was 0.2 mL/min, and the column temperature was maintained at 30 °C. The injection volume was 1.0 µL. The detection wavelength was set at 254 nm.

The optimized MS conditions were as follows: capillary voltage was 3800 V in the negative mode; dry gas (N_2_) flow rate was 6.0 L/min; nebulizer gas (N_2_) pressure was 0.8 bar; and dry gas temperature was set at 200 °C. The scan range was 50 to 800 Da. Sodium formate solution with a concentration of 10 mM was used as an internal calibration solution. Auto MS/MS mode was selected to collect the secondary MS data by applying different CEs with a collision gas (high-purity argon) after choosing the precursor ions. The Collision Cell RF voltage was set at 150.0 Vpp.

### 3.4. Data Processing

The Q-TOF-MS raw data files were first converted into the analysis base file (ABF) format by Abf Converter (version 4.0.0) and then further processed by MS-DIAL (version 4.20) [49]. Data processing included peak extraction time (0.25–45.0 min), data collection (mass scan range of 50–800 Da), MS1 mass tolerance (0.01 Da), MS2 mass tolerance (0.025 Da), peak detection, deconvolution, filtering (the peak count filter was set at 14.3%), peak alignment, and integration. A set of three-dimensional data matrices composed of RT, the mass-to-charge ratio (*m*/*z*), and peak intensity were generated.

### 3.5. Assay of DPPH Free Radical Scavenging Activity

DPPH radical scavenging activity was evaluated according to the method described by Sarikurkcu with slight modifications [50]. Briefly, 100 μL of the sample solution with different concentrations was mixed with 100 μL of DPPH solution (0.2 mM). The absorbance of the mixture was measured at 517 nm after 30 min of incubation in the dark at room temperature. L-ascorbic acid was used as the reference compound. The test was repeated three times, and the average value was calculated. The scavenging rate (%) of DPPH free radicals was calculated according to a formula:The scavenging rate (%) = [1 − (A*sample* − A*blank*)/A*control*] × 100%,(1)
where A*sample* was the absorbance of 100 μL of sample solution and 100 μL of DPPH solution; A*blank* was the absorbance of 100 μL of sample solution and 100 μL of ethanol; and A*control* was the absorbance of 100 μL of ethanol and 100 μL of DPPH solution.

### 3.6. Determination of Total Phenolic Content

The total phenolic content of the extracts was analyzed using the Folin–Ciocalteu method described by Sarikurkcu with slight modifications [50]. Briefly, 5 mL of the sample solution (1.0 mg/mL of the extract) was mixed with 60 mL of ultrapure water, and then 1 mL of Folin–Ciocalteu reagent was added. The mixture was incubated for 3 min, and then 5 mL of a 10% (*w*/*v*) sodium carbonate (Na_2_CO_3_) solution was added. The volume was set with ultrapure water at 100 mL. The final mixture was incubated for 120 min at 25 °C, and absorbance was measured at 765 nm using a spectrophotometer (UV-2550 UV-VIS, Shimadzu, Japan). The results of total phenolic content were expressed using a standard curve of gallic acid.

### 3.7. Statistical Analysis

The processed data were introduced into SIMCA 14.1 (Umetrics, Umeå, Sweden) for multivariate statistical analyses. All variables were pareto-scaled prior to chemometric analysis. PCA was utilized to analyze the degree of correlation of the data according to the aggregation of each batch of mulberry leaves in the group and the dispersion of different batches of mulberry leaves outside the group. Meanwhile, OPLS-DA was utilized to study the differences between the different samples, mainly for pairwise comparison. The differential components were selected according to the variable importance in projection (VIP > 5.0) obtained from the OPLS-DA model and *p*-values (*p* < 0.05) calculated by the Mann–Whitney U test. The processed data were subjected to log2 transformation. The volcano plot was obtained by using the R-4.0 language.

## 4. Conclusions

The chemical profiling of mulberry leaves harvested in different months was systematically investigated by UPLC–Q-TOF-MS. A total of 77 compounds were detected, of which 65 were tentatively identified. Flavonoid glycosides and phenolic compounds were found to be the main compounds in mulberry leaf extracts. Moreover, batch A, harvested in December, exhibited the strongest radical-scavenging activity. In contrast, batch B, harvested in March, exhibited the weakest radical-scavenging activity. It was speculated that the radical-scavenging activity was related to the polyphenols. The results obtained in this study might contribute to further investigation of mulberry leaves in terms of their potential application as food.

## Figures and Tables

**Figure 1 molecules-28-01947-f001:**
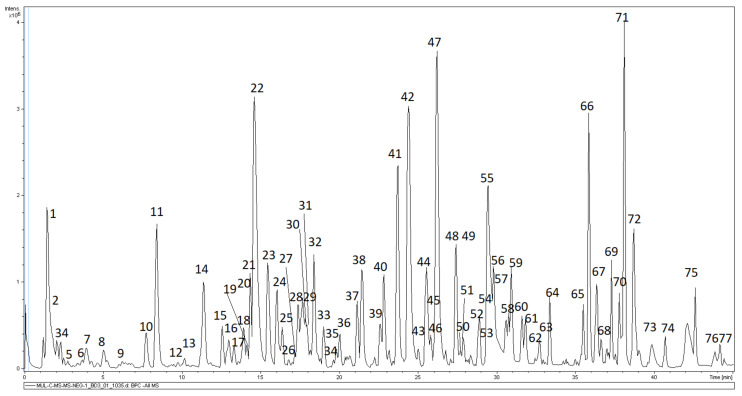
The BPC of mulberry leaf extracts.

**Figure 2 molecules-28-01947-f002:**
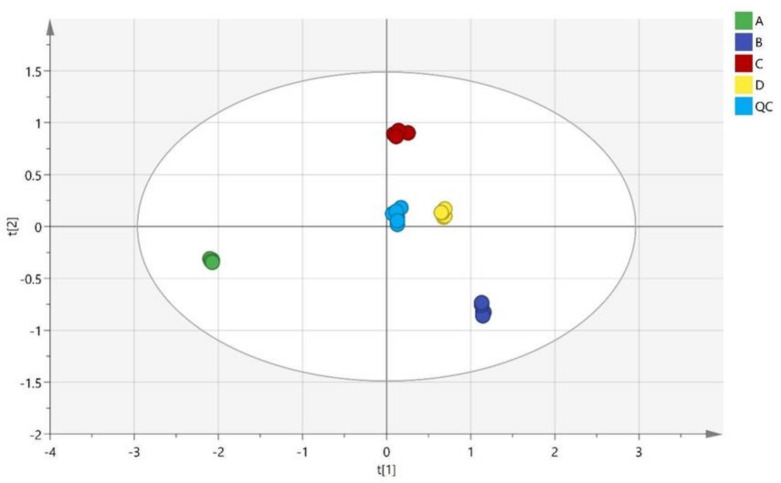
PCA scores of mulberry leaves harvested in different months.

**Figure 3 molecules-28-01947-f003:**
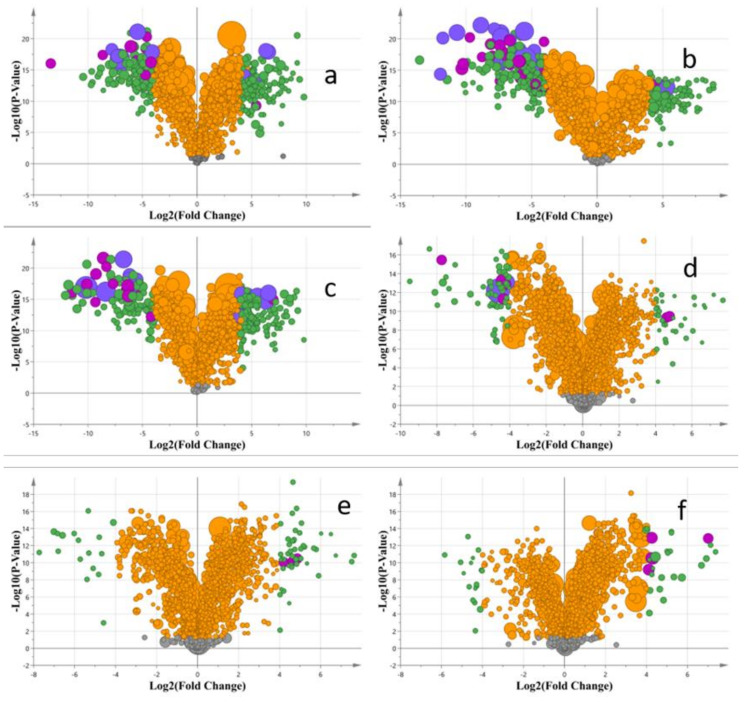
The volcano plots with different batches of mulberry leaves. ((**a**): A&B; (**b**): A&C; (**c**): A&D; (**d**): B&C; (**e**): B&D; (**f**): C&D).

**Figure 4 molecules-28-01947-f004:**
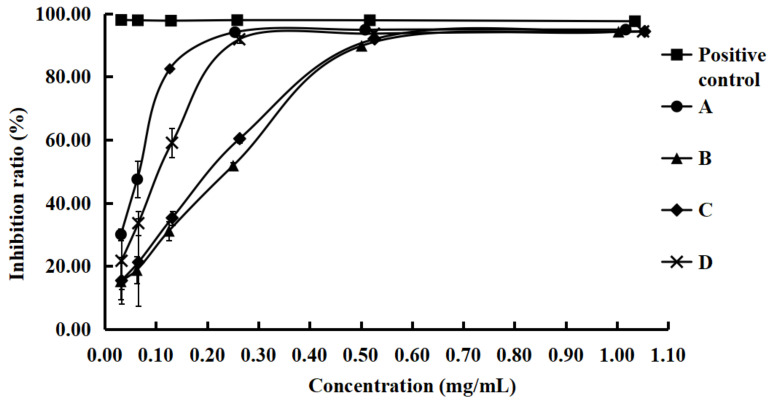
The results of the free radical-scavenging activity of mulberry leaf extracts.

**Table 1 molecules-28-01947-t001:** Compounds identified from the 75% ethanol extracts of mulberry leaves by UPLC–Q-TOF-MS.

No.	RT (min)	Measured *m/z*	[M − H]^−^	Theoretical *m/z*	Error (ppm)	Fragments	Identification
Organic acids and vitamin derivatives							
1	1.6	191.0561	C_7_H_11_O_6_	191.0561	0.1	137,127	Quinic acid [27]
2	1.7	133.0143	C_4_H_5_O_5_	133.0142	−0.7	/	L-Malic acid [27]
3	2.1	337.0781	C_12_H_17_O_11_	337.0776	−1.3	277,174,157,114	L-Ascorbic acid glucoside [28]
4	2.3	191.0202	C_6_H_7_O_7_	191.0197	−2.3	111	Citric acid [28]
5	2.8	292.1407	C_12_H_22_NO_7_	292.1402	−1.8	130	N-Fructosyl isoleucine (MS DIAL)
6	3.7	282.0844	C_10_H_12_N_5_O_5_	282.0844	−2.2	150,133	Guanosine (MS DIAL)
Phenolic compounds							
7	4.0	331.0679	C_13_H_15_O_10_	331.0671	−2.6	169,151,125	Gallic acid glucoside
8	5.0	315.0724	C_13_H_15_O_9_	315.0722	−0.8	153	Protocatechuic acid glucoside isomer 1 [28]
9	6.3	153.0199	C_7_H_5_O_4_	153.0193	1.9	/	Protocatechuic acid [29]
10	7.8	315.0730	C_13_H_15_O_9_	315.0722	0.3	153,152,135,109	Protocatechuic acid glucoside [28]
11	8.4	315.0720	C_13_H_15_O_9_	315.0722	0.4	153,152	Protocatechuic acid glucoside isomer 2 [28]
12	9.8	515.1412	C_22_H_27_O_14_	515.1406	−0.4	312,311,221,179,135	Dicaffeoylquinic acid [30]
13	10.2	359.0990	C_15_H_19_O_10_	359.0984	−1.6	197,179,166,153,135,123	Syringic acid hexoside [31]
14	11.4	353.0881	C_16_H_17_O_9_	353.0878	−0.9	191,179,135	3-*O*-Caffeoylquinic acid [29]
15	12.6	339.0733	C_15_H_15_O_9_	339.0722	−3.4	177	Aesculin [29]
16	12.9	299.0772	C_13_H_15_O_8_	299.0772	0.1	137	Hydroxybenzoyl hexoside [31]
17	13.3	515.1412	C_22_H_27_O_14_	515.1406	−1.1	324,323, 191,161	3,5-dicaffeoylquinic acid [32]
18	14.0	339.0731	C_15_H_15_O_9_	339.0722	−2.9	177	Aesculin isomer 1 [28]
22	14.6	353.0883	C_16_H_17_O_9_	353.0878	−1.5	191	Chlorogenic acid ^a^
23	15.5	353.0878	C_16_H_17_O_9_	353.0878	0.0	191,179,173,135	5-*O*-Caffeoylquinic acid [29]
64	33.3	315.1246	C_18_H_19_O_5_	315.1238	−2.5	175,163,160,148,135	Protocatechuic acid hexoside [33]
Flavonoids							
19	14.2	465.1050	C_21_H_21_O_12_	465.1038	−2.5	343,303,299,286,285,275,181,179,177,153,151,125	Taxifolin-*O*-glucoside [32]
21	14.4	639.2878	C_28_H_47_O_16_	639.2870	−1.3	550,549,387,179,161,149,119	Quercetin *C*-hexoside glucuronide [33]
24	16.1	625.1407	C_27_H_29_O_17_	625.1410	0.5	464,463,462,301,299	Quercetin hexosylhexoside [29]
25	16.4	431.1928	C_20_H_31_O_10_	431.1923	−1.3	387,315,297,153,152	Apigenin hexoside [33]
27	17.2	449.1098	C_21_H_21_O_11_	449.1089	−1.9	287,269,260,259,179,151,125	Cyanidin hexoside [29]
28	17.4	431.1931	C_20_H_31_O_10_	431.1923	−1.9	153,152	Apigenin *C*-glucoside [29]
30	17.9	611.1614	C_27_H_31_O_16_	611.1618	0.5	241	Taxifolin-*O*-rutinoside [32]
31	18.0	711.1422	C_30_H_31_O_20_	711.1414	−1.1	668,667,505,463,462,301,299	Quercetin malonyl-dihexoside [27]
32	18.4	609.1462	C_27_H_29_O_16_	609.1461	−0.1	448,447,446,286,285,284,283	Kaempferol hexosylhexoside [29]
34	19.7	405.1200	C_20_H_21_O_9_	405.1191	−2.2	243,225,201,199,175	2,3,5,4′-Tetrahydroxystilbene-2-*O*-β-D-glucoside [34]
37	21.1	625.1411	C_27_H_29_O_17_	625.1410	−0.2	301,300	Quercetin di-*O*-glucoside [28]
38	21.5	463.0895	C_21_H_19_O_12_	463.0882	−2.7	302,301,300,151	Quercetin 3-*O*-hexoside [29]
39	22.6	285.0772	C_16_H_13_O_5_	285.0768	−1.4	268,267,255,225,213,211,187,183,171	Kaempferol ^a^
40	22.9	609.1472	C_27_H_29_O_16_	609.1461	−1.8	285,284	Kaempferol 3-*O*-sophoroside [35]
41	23.7	609.1473	C_27_H_29_O_16_	609.1461	−1.9	302,301,300	Rutin ^a^
42	24.4	463.0884	C_21_H_19_O_12_	463.0882	−0.3	301,300	Isoquercitrin ^a^ [28]
43	25.6	549.0897	C_24_H_21_O_15_	549.0886	−2.0	301,300	Quercetin 3-*O*-malonyl-glucoside [27]
44	25.6	505.0998	C_23_H_21_O_13_	505.0988	−2.1	301,300	Quercetin-3-*O*-glucosyl-6″-acetate [28]
45	25.5	593.1523	C_27_H_29_O_15_	593.1512	−1.9	286,285,284	Kaempferol-3-*O*-rutinoside [28]
47	26.2	447.0942	C_21_H_19_O_11_	447.0933	−2.1	285,284,255,227	Kaempferol-3-*O*-glucoside [28]
48	27.4	489.1042	C_23_H_21_O_12_	489.1038	−0.6	286,285,284	Kaempferol-3-*O*-6″-*O*-acetyl-B-D-glucopyranoside [28]
49	27.4	533.0940	C_24_H_21_O_14_	533.0937	−0.6	286,285,284	Kaempferol-malonyl-glucoside [30]
50	27.7	489.1048	C_23_H_21_O_12_	489.1038	−1.9	285,284	Kaempferol-acetyl-glucoside [30]
51	27.9	521.1313	C_24_H_25_O_13_	521.1301	−2.4	353,315,223,205,191,190,153,152	Quercetagenin acetyl hexoside [33]
52	28.8	477.1781	C_24_H_29_O_10_	477.1766	−3.1	316,315,180,179,165,161,153,149,135	Isorhamnetin 3-*O*-hexoside [33]
53	28.8	523.1838	C_25_H_31_O_12_	523.1821	−3.2	316,315,179,165,161,153	Ligustroside [36]
54	30.4	301.0355	C_15_H_9_O_7_	301.0354	−0.3	151,121	Quercetin [28]
55	29.3	463.1620	C_23_H_27_O_10_	463.1610	−2.2	300,194,193	Quercetin-*O*-hexoside [33]
56	29.8	477.1773	C_24_H_29_O_10_	477.1766	−1.3	315,193,179,135	Quercetin glucuronide [33]
58	30.3	519.1870	C_26_H_31_O_11_	519.1872	0.3	317,316,315,193,179,175,165,161,153,149,135	Isorhamnetin acetyl hexoside [33]
60	30.9	519.1890	C_26_H_31_O_11_	519.1872	−3.6	310,309,307,297,193,135	Matairesinoside [37]
Fatty acid derivatives							
61	31.8	327.2168	C_18_H_31_O_5_	327.2177	2.7	229,211,183,171	Trihydroxy-octadecadienoic acid [28]
63	32.6	329.2344	C_18_H_33_O_5_	329.2333	−3.3	267,256,255,229,213,211,187,183,171,139	Trihydroxy-octadecenoic acid [28]
65	35.3	309.2055	C_18_H_29_O_4_	309.2071	5.4	171,137	Linolenic acid hydroperoxide isomer 1 [28]
70	37.8	293.2126	C_18_H_29_O_3_	293.2122	−1.4	276,275,235,183,172,171,121	Hydroxy-octadecatrienoic acid [28]
75	42.6	277.2173	C_18_H_29_O_2_	277.2173	−0.1	/	Linolenic acid [28]
77	44.2	279.2330	C_18_H_31_O_2_	279.2330	−0.3	/	Linoleic acid [28]
Quinonoids							
66	35.8	309.1138	C_19_H_17_O_4_	309.1132	−2.0	286,254,209	Tanshinone IIB [38]
68	36.6	307.0977	C_19_H_15_O_4_	307.0976	−0.5	289,279,277,265,263,261,248,247,224,223,157	Tanshinoldehyde [39]
Terpenoids							
67	36.3	339.1237	C_20_H_19_O_5_	339.1238	0.3	307,292,291,203,199,177,161,135,122	8-Prenylnaringenin [40]
69	37.3	339.1610	C_21_H_23_O_4_	339.1602	−2.3	204,203,177,149,148,134	6-Prenylnaringenin [41]
71	38.1	339.1613	C_21_H_23_O_4_	339.1602	−3.3	217,159,147,135	A novel terpenoid-type phytoalexin [42]
Terpene							
73	39.8	571.2899	C_32_H_43_O_9_	571.2913	2.3	391,315,283,256,255,241,152	Ganoderic acid H [43]
Lignan							
74	40.7	353.1771	C_22_H_25_O_4_	353.1758	−3.6	218,217,202,159,149,147,134	Variegat C [44]
Unknown							
20	14.4	549.2557	C_25_H_41_O_13_	549.2553	−0.7	339	n.a. ^b^
26	16.8	399.1308	C_18_H_23_O_10_	399.1297	−2.7	237,220,219,193,175,63	n.a.
29	17.7	433.2079	C_20_H_33_O_10_	433.2079	0.1	387,225,207,189,163,161,153,152,123	n.a.
33	19.0	579.2656	C_26_H_43_O_14_	579.2658	0.4	534,533,369,179,161,149,143,131,119,113	n.a.
35	19.9	579.2668	C_26_H_43_O_14_	579.2658	−1.6	313,179,161,149,143,131,119,113	n.a.
36	20.0	533.2614	C_25_H_41_O_12_	533.2604	−1.9	195	n.a.
46	25.8	579.2083	C_28_H_35_O_13_	579.2096	3.8	417,402,181	n.a.
57	29.8	523.1834	C_25_H_31_O_12_	523.1821	−3.2	316,315,193,135	n.a.
59	30.9	477.1782	C_24_H_29_O_10_	477.1766	−3.3	298,297,135	n.a.
62	32.7	227.1296	C_12_H_19_O_4_	227.1289	−3.3	183	n.a.
72	38.7	647.2305	C_39_H_35_O_9_	647.2287	−2.9	469,360,359,241,227,177	n.a.
76	43.9	621.4376	C_36_H_61_O_8_	621.4372	−0.6	311	n.a.

^a^ Compounds were identified by the standards. ^b^ n.a., compounds were not available.

**Table 2 molecules-28-01947-t002:** Differential compounds of mulberry leaves in different batches.

No.	RT (min)	Compound	A&B	A&C	A&D	B&C	B&D	C&D
40	22.9	Kaempferol 3-*O*-sophoroside	++					
32	18.4	Kaempferol hexosylhexoside	++					
69	37.3	6-Prenylnaringenin	++	++				
66	35.8	Tanshinone II_B_	--		+			
72	38.7	n.a.	++	++	++		-	
18	14.0	Aesculin isomer 1	+	+	+			
37	21.1	Quercetin di-*O*-glucoside	+		++			
30	17.9	Taxifolin-*O*-rutinoside	+	+	+			
60	30.9	Matairesinoside		++				
59	30.9	n.a.		++				
38	21.5	Quercetin 3-*O*-hexoside		++				
64	33.3	Protocatechuic acid hexoside		++				
58	30.3	Isorhamnetin acetyl hexoside		++	-			
65	35.3	Linolenic acid hydroperoxide isomer 1		--	--			
39	22.6	Kaempferol		+				
8	5.0	Protocatechuic acid glucoside isomer 1		+				
51	27.9	Quercetagenin acetyl hexoside		+	-			
53	28.8	Ligustroside		+	-			
22	14.6	Chlorogenic acid						-
31	18.0	Quercetin malonyl-dihexoside						-
41	23.7	Rutin				++		
1	1.6	Quinic acid				++		
63	32.6	Trihydroxy-octadecenoic acid				-		
40	22.9	Kaempferol 3-*O*-sophoroside			++			
10	7.8	Protocatechuic acid glucoside	+		+			

Note: “+” indicated that the content of the differential compound in the previous batch was more than that in the latter. “-” was the opposite. The more plus signs showed the greater the difference in the content.

**Table 3 molecules-28-01947-t003:** The harvest time of mulberry leaves.

Batch	Harvest Time	Average Temperature and Precipitation
A	December 2019	2–11 °C and 46 mm
B	March 2020	6–14 °C and 132 mm
C	April 2020	11–20 °C and 107 mm
D	May 2020	17–26 °C and 120 mm

## Data Availability

The data that support the findings of this study are available from the corresponding author upon reasonable request.

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
