# Peer review of "Investigation on Antioxidant Activity and Different Metabolites of Mulberry (Morus spp.) Leaves Depending on the Harvest Months by UPLC–Q-TOF-MS with Multivariate Tools"

_molecules, 2023, doi:10.3390/molecules28041947_

Round 1

Reviewer 1 Report

I have several comments on this manuscript:

1. The authors should get some help in English writing as I observed many grammatical errors in the manuscript.

2. Experimental design: not so interesting and missing rationalization.  There are several problems with the experimental design: sample collection and analysis.  Why chose batch B for UHLC analysis?  Shouldn't the authors compare the results from the 4 sets?  Why harvested the leaves over those 4 periods of time?  Why not collect the leave every month or two for comparison?  Why chose to perform DPPH assay but not other antioxidant assays and where are the total phenolic results?

3. DPPH assay results should be expressed as the IC50.

4. The content under discussion should be within the results and some of them should be materials and methods.

Reviewer 2 Report

Your work is valuable, well constructed and I believe it will be better with some suggestions. You can find my recommendations below:

(i) Information about the advantages and advantages of the device  mentioned in the title part of the study can be given in the introduction.

(ii) There are some errors about abbreviations in the study, they should be corrected.

It should be clearly stated why ultrasonic sonication and related solvent are used. for example, why did you prefer maceration and soxlet techniques? And why didn' you used other extraction solutions?

Round 2

Reviewer 1 Report

The revision was better than the original submission.  The initial submission did not refer to the supplementary materials at all and the clarity in writing was very low.  I have the following comments:

1. I strongly recommend that the authors have a native English speaker read the manuscript.  There are still many incorrect sentence structures and grammatical errors shown. 

2. The authors should include other antioxidant assays (e.g. FRAP and ABTS) in the report as well. 

3. Lines 131-142 should be moved to the results

4. Lines 144-159 should be moved to the results

5. There is no need to reiterate the information presented in a table in the text (lines 161-213) as this creates redundancy.  Instead, the author should summarize the data differentaly, such as in a form of a diagram or chart.
